# Personal approach for cancer treatment: A meta-analysis of Phase II clinical trials

**Mikhail B. Potievskiy**[1]*, **Elena P. Zharova**[1], **Lidia A. Nekrasova**[1], **Airat I. Garifullin**[1], **Ivan V. Korobov**[1], **Nikita E. Shevchenko**[1], **Anastasia A. Zabolotneva**[2], **Dmitriy N. Atochin**[3,4], **Andrei D. Kaprin**[1,5], **Peter V. Shegai**[1]

**1** FSBI "National Medical Research Radiological Center" of the Ministry of Health of the Russian Federation (FSBI "NMRRC"), Obninsk, Russia, **2** Department of Biochemistry and Molecular Biology, N.I. Pirogov Russian National Research Medical University, Moscow, Russia, **3** Cardiovascular Research Center, Department of Medicine, Massachusetts General Hospital, Charlestown, Massachusetts, United States of America, **4** Department of Psychiatry, Boston VA Medical Center West Roxbury, Veterans Affairs Boston Healthcare System and Harvard Medical School, Boston, Massachusetts, United States of America, **5** Peoples' Friendship University of Russia—RUDN University, Moscow, Russia

* potievskiymikhail@gmail.com

## Abstract

### Background

To date, no meta-analysis has studied the general outcomes of personalized cancer drug therapy with a focus on current targeted, immunotherapy, and multi-agent phase II clinical trials.

### Objective

We conducted a systematic review and meta-analysis to provide a comprehensive overview of outcomes in patients undergoing personalized genomics-based versus non-personalized treatment in oncology.

### Data source

We searched for publications in PubMed, dedicated to specific cancer drug treatment and phase II clinical trials, and published from 2010 to 2021. The search dates were from 20.10.23 to 20.11.23. The final data check was on 20.12.23.

### Selection criteria

Studies of chemotherapy, immunotherapy, and targeted therapy were included. Only trials, including adults (more than 18 y.o.) were selected. The personalization was evaluated based on genetic markers and study design.

**Data availability statement:** All data files are available from the zenodo database (accession number 10966214 (https://www.doi.org/10.5281/zenodo.10966214)). All the code is available on github: https://github.com/MikhailPot/PreciseOnco_meta-analysis.

**Funding:** The author(s) received no specific funding for this work.

**Competing interests:** The authors have declared that no competing interests exist.

## Data collection and analysis

The study was performed following PRISMA guidelines. Two reviewers worked independently to select studies and arms, and one checked the results. Three reviewers extracted the data, and another reviewer independently checked it. The proportional meta-analysis, random-effects model, and meta-regression were employed to evaluate the effects of genomics-based personalization and other study design parameters (randomization, multi-central protocol, pre-treatment, therapy type, number of patients per arm, and journal impact factor) on the treatment outcomes. Mann-Whitney was employed to compare survival medians, $p < 0.05$.

## Results

We evaluated 50 studies, having 81 arms and 6536 patients. Response Rate (RR) and 1-year Progression-Free Survival (PFS) were significantly higher in personalized arms ($p = 0.009$ and $p = 0.011$). Medians of PFS and Overall Survival (OS) were also higher in personalized arms ($p = 0.018$ and $p = 0.032$). Proportional meta-analysis and meta-regression detected a significant positive association between personalized treatment and RR ($p = 0.037$). The same results were obtained for 1-year PFS and OS rates ($p = 0.043$ and $p = 0.022$). Previous drug treatment, type of therapy (targeted therapy/immunotherapy/cytotoxic chemotherapy), study design, and journal impact factor did not affect RR, PFS, and OS. Personalization only affected the treatment outcomes.

## Conclusion

This study discovered the benefits of a personal approach to cancer treatment using genomic data. The personalized approach improved cancer outcomes and offers promising therapeutic potential for the further development of cancer treatment.

## Registration

PROSPERO record ID CRD42024504021

## Introduction

Precision medicine continues to revolutionize oncology by personalizing treatment strategies, based on the genetic and molecular characteristics of the tumor and patient. Advances in technology, such as next-generation sequencing and data analysis, are contributing to the increasing implementation of precision medicine in clinical practice [1,2]. By identifying specific genome data, and genetic mutations, and guided by other patient characteristics precision medicine allows for more effective treatments with fewer side effects [3]. In the long term, precision medicine approaches demonstrate the potential to reduce healthcare costs and increase understanding of disease mechanisms [4,5].

One of the successes in precision oncology is trastuzumab deruxtecan, targeting HER2-positive breast cancer with remarkable specificity and efficacy [6]. This

targeted therapy demonstrates the transformative potential of precision medicine in selecting treatment regimens for specific patient populations. PARP inhibitors that suppress poly-ADP ribose polymerase (PARP) are increasingly used in patients with inherited BRCA1 or BRCA2 mutations in various cancers, including colorectal, breast, and ovarian cancers [7,8]. The study of biomarkers is integral to the diagnosis [9–12]. For example, detecting IDH1/IDH2 mutation and MGMT promoter methylation status is essential for diagnosis, treatment selection, and determining prognosis in glioma patients [9]. In melanoma, personalized therapy has shown significant efficacy in patients with BRAF V600E mutations, resulting in better response rates and survival outcomes [10]. Personalized therapy in colorectal cancer includes the use of targeted therapies such as cetuximab and panitumumab for patients with RAS mutations [11]. The application of personalized therapy in leukemia involves targeted treatments like imatinib for patients with the BCR-ABL fusion gene [4,12].

The successful application of precision medicine in clinical oncology presents various challenges. One of them is the limited availability of targeted drugs for the treatment of patients with certain genetic mutations in low-income countries [13,14]. The widespread practice of precision medicine is hampered by problems such as the interpretation of molecular data in the absence of standardized guidelines for the implementation of precision medicine in clinical practice [15]. This is why more rigorous evidence of patient benefit from precision medicine approaches is necessary. Although there is much evidence of the benefits of precision approaches within certain diseases [16,17], the evidence for the benefits of precision approaches in clinical oncology is not fully evaluated. Most studies focus on the outcomes of personalization for certain cancer treatments [15] with specific drugs [18]. At the same time, the global impact of treatment personalization is not highlighted in the literature [19].

Precision therapy in clinical oncology is a cutting-edge approach that has shown promising results in recent studies [20]. Despite its relevance, a limited number of meta-analyses focus on this topic in the current literature. The last wide meta-analysis (Schwaederle M. et al 2015) dedicated to the impact of a personal approach in cancer drug treatment, included only studies from 2010–2012 [4,21].

Our meta-analysis aims to provide a comprehensive overview of outcomes in cancer patients undergoing personalized genomics-based therapy versus patients with non-personalized treatment. By synthesizing data from a plethora of studies, we aimed to offer a detailed evaluation of the efficacy of precision therapy in cancer care. Our objective was to compare response rates (RR), overall survival (OS), and progression-free survival (PFS) between personalized and non-personalized studies.

## Methods

The meta-analysis and systematic review, including the data searches, study selection, data extraction, and categorization with further statistical analysis were performed in accordance with PRISMA (Preferred Reporting Items for Systematic Review and Meta-Analysis) guidelines [22–24]. The study protocol was registered in PROSPERO (International Prospective Register of Systematic Reviews, https://www.crd.york.ac.uk/PROSPERO), record ID CRD42024504021.

### Search strategy

We searched for publications in PubMed (https://PubMed.ncbi.nlm.nih.gov/), the search dates were from 20.10.23 to 20.11.23. The searches were based on MeSH terms, article title, and publication type: (("Clinical Trials, Phase II as Topic"[Mesh] AND cancer*[tiab])) NOT (review*[pt] OR Meta-analysis[pt] OR Systematic review[pt]), publication dates from 2010 to 2021, "studies in humans" was selected as an additional filter. The searches were re-run before the final analysis. The final data check was on 20.12.23.

### Eligibility criteria and study selection strategy

We included studies dedicated to specific cancer drug treatments and phase II clinical trials. Studies of chemotherapy, immunotherapy, targeted therapy, and multi-agent trials, including various treatments, were selected. Only trials, including adults (more than 18 y.o.) were selected.

Non-inclusion/Exclusion criteria were study design different from the phase II clinical trial, participants under 18 and more than 85 y.o., absence of any specific cancer drug treatment, and no detailed information on patient inclusion parameters. Studies without optimal endpoints such as response rate (RR), progression-free survival (PFS), or overall survival (OS) were excluded. Incomplete statistical analysis was also an exclusion criterion. We included only articles with English full text. Unpublished studies were not included. Any type of reviews and case reports were also excluded.

The personalization was evaluated based on genetic markers, available additional information about the studied parameters, and the design of the trial. We considered only studies based on genomic patient data as personalized genomics-based, while other studies were considered non-personalized. During the selection, we evaluated each study arm independently. If it was impossible to evaluate the personalization, the studies or its arm was excluded. If personal and non-personal approaches were simultaneously detected in the same arm, the arm was excluded.

### Assessment of study quality

The included study quality was independently assessed using the Cochrane Risk of Bias tool [25], the results are presented in S2 Table and the online repository (https://www.doi.org/10.5281/zenodo.10966214). Each arm was considered having the same risk of bias as its study. The assessment was conducted by two investigators independently.

### Data extraction and categorization

After the searches were completed, two reviewers worked independently to select studies and arms, applying the eligibility criteria, and one reviewer checked the results of their work after the selection. Before the first step of the selection was finally done, the researchers were blinded to each other's decisions. All disagreements between individual judgments were resolved in argument discussion with the moderation of the principal investigator [23,24,26]. The decisions were recorded in Excel. The studying parameters were the number of patients, previous treatment, study design (randomization, single or multi-central protocol), type of therapy (chemotherapy/immunotherapy/targeted therapy), journal impact factor, and the outcome endpoints, such as response rate, overall and progression-free survival rates (medians, total survival rates, and time of the observation).

In the next step, three reviewers collected the data, and one reviewer checked the results. The missing data was requested from the investigators. If the missing data were not received, only available data was included. If there were no crucial parameters for study quality check, design, and inclusion parameters were unclear, the study was excluded. Each case of missing data was discussed by reviewers. After an independent evaluation, we selected studies dedicated to personalized treatment and studies for the comparison group, without personalization. If a personal approach was used in the study in the comparison group, and standard treatment was used in the control group, the arms were evaluated in different groups [23,24].

### Data synthesis and statistics

We synthesized all the study parameters, including the number of patients, previous treatment, study design, therapy type (chemotherapy/immunotherapy/targeted therapy), journal impact factor, response rate, and 1-year and medians OS and PFS. The 1-year survival rates were calculated from survival median and observation time or total survival rates and observation time. We considered multi-agent therapy as targeted or immune-oncological treatment if those were appointed in combination with other treatments.

We performed proportional meta-analysis using the RStudio program, utilizing the R programming language and packages "meta", "metafor," and "tidyverse" [27]. We calculated relative risks using the Der Simonian-Laird algorithm [28]. Heterogeneity was evaluated based on the $\tau^2$-statistic [28,29] and $I^2$ Higgins-Thompson statistic [30,31]. $\tau^2$ confidence intervals were calculated with the Jackson method [32]. We compared the effects using the Chi-square test for categorical variables and employed the t-test or Mann-Whitney test based on the type of distribution (normal or non-normal) for

continuous variables (RR, OS, and PFS), with a significance level set at p<0.05. The distribution normality was tested with the Shapiro-Wilk test, α=0.05. Taking into account the heterogeneity, we utilized the random-effects model [33]. We conducted meta-regression to assess the impact of the personal approach implementation and other parameters at p<0.05, 95%. CIs and p-values were estimated with the Knapp-Hartung method [34,35]. Each subgroup included a minimum of 10 studies.

The risk of bias was evaluated with the GRADE (Grading of Recommendations Assessment, Development, and Evaluation) approach [36] to assess the validity of the evidence of RR, OS, and PFS analysis (S3 Table). The analysis included the study of inconsistencies in results, trial design, risk of bias, indirectness of evidence, and publication bias. Two investigators conducted the assessment independently.

## Results

Initially, we selected 803 studies, based on Phase II clinical trial results for the analysis (Fig 1). 451 trials did not meet the inclusion criteria after title and abstract evaluation. In the result of the full-text analysis, 295 studies were fully excluded, and 7 studies were partly excluded: 18 unsuitable arms were removed from the final dataset. Thus, we selected 50 studies, having 81 arms, and a total patient number of 6536. In 35 arms (43.2%), including 2275 patients (34.8%) the treatment was personalized with a genomics-based approach, and in 46 arms (56.8%), including 4261 patients was non-personalized (65.2%). The parameters distribution is presented in Table 1. 6 arms (7.4%) had high risk of bias, 20 (24.7%) – moderate, and 55 (67.9%) – low (S2 Table).

We compared the effects of personal genomics-based and non-personal approaches in cancer drug treatment. RR and 1-year PFS were significantly higher in personalized arms, p=0.009 and p=0.011 (Table 2). In the case of a 1-year OS, there was no significant difference between personalized and non-personalized arms.

The between-study heterogeneity variance was quite high for RR, PFS, and OS (Fig 2). The $\tau^2$ and Higgins-Thomson statistics were estimated as $\tau^2=0.028$ [0.023; 0.066] with $I^2=88.7\%$ [85.6%; 91.1%] for RR, $\tau^2=0.091$ [0.074; 0.176] with

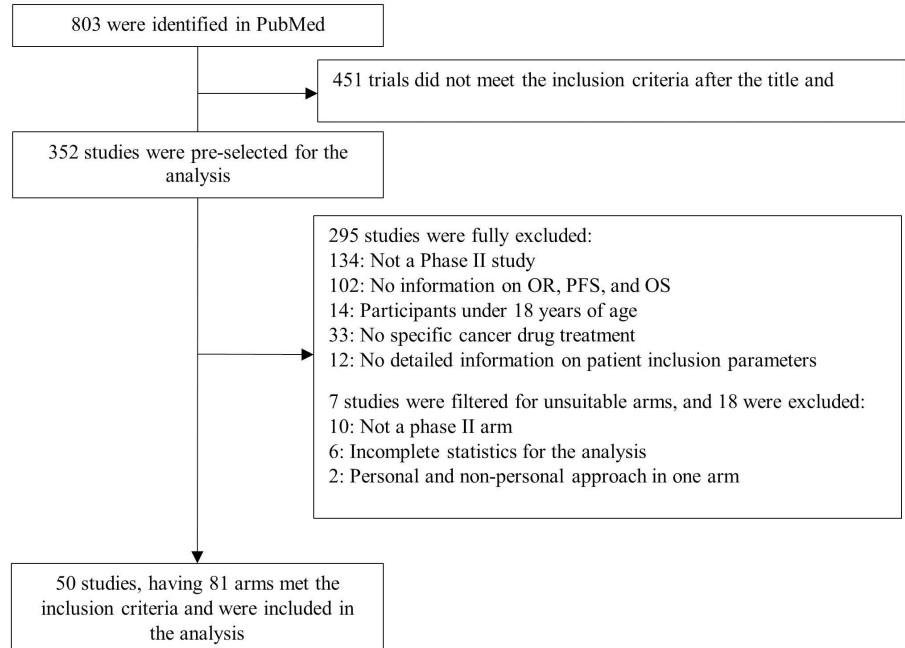

**Fig 1. Workflow of study selection (PRISMA data extraction diagram).**

**Table 1. Arms and parameters, N(%).**

| Parameters \ Statistics | All arms evaluation (n=81) | P-value | Non-personal (n=46) | Personal (n=35) | P-value |
|---|---|---|---|---|---|
| Randomization | 52 (64.2) | 0.011* | 22 (47.8) | 22 (62.9) | 0.368 |
| Multi-central protocol | 14 (17.3) | $3.89*10^{-8}$* | 8 (17.4) | 6 (17.1) | 0.521 |
| Pre-treated | 36 (44.4) | 0.317 | 18 (39.1) | 16 (45.7) | 0.292 |
| Targeted/Immunotherapy | 26 (32.1) | $1.456*10^{-8}$* | 16 (34.8) | 13 (37.1) | 0.768 |
| Chemotherapy | 55 (67.9) | | 30 (65.2) | 22 (62.9) | |
| Number of patients per arm>40 | 60 (74.1) | $1.406*10^{-6}$* | 23 (50) | 26 (74.3) | 0.585 |
| Journal 5 year IF > 10 | 24 (29.6) | $2.457*10^{-4}$* | 12 (26) | 10 (28.6) | 0.326 |

*Significant difference, chi-square test, $p<0.05$. Targeted and immunotherapeutic (including multi-agent) arms were compared to arms, where only cytotoxic chemotherapy was appointed. Non-personal and personal arms were compared with each other (subgroup analysis).

$I^2=96.7\%$ [96.2%; 97.1%] for PFS and $\tau^2=0.15$ [0.101; 0.22] with $I^2=97.8\%$ [97.6%; 98.1%] for OS. High heterogeneity rates may be associated with different response absolute metrics of outcomes for various cancers, disease stages and treatments.

Proportional meta-analysis and meta-regression detected a significant association between personal genomics-based cancer drug treatment and RR. RRs were significantly higher in personalized arms (Fig 2a, Table 2), p=0.037. The same results were obtained for 1-year PFS and OS rates (Fig 2b, 2c), p=0.043 and 0.022. In the case of PFS, we detected significantly worse outcomes in pre-treated patients. For example, the results of the second and later line of treatment, p=0.027, may be associated with metastatic cancer and advanced disease stages. Lower PFS was associated with a lower number of patients per arm, p=0.024. This finding may be based on limitations of the study, due to high heterogeneity rates and diverse sampling approaches in different studies [30]. However, such results were not obtained for OS and RR. We conducted multivariable meta-regression for these PFS significant parameters. In the result, the main PFS prognostic factor was the previous treatment (Table 3).

The same results were obtained in PFS and OS medians analysis. We evaluated the median difference between personal and non-personal groups. Median PFS was significantly higher in personalized studies, p=0.015, similar to OS, p=0.032. Meta-regression is a more powerful method; however, it could not be employed for meta-analysis of the difference of medians without additional information or control groups in the studies.

### Subgroup analysis

We conducted a subgroup analysis in personalized and non-personalized studies. There was no difference in studying parameters between personalized and non-personalized studies (Table 1). Therefore, the inclusion of genomic markers is suggested as the main factor improving RR, OS, and PFS in the result of cancer drug treatment, based on phase II clinical trials. Study design (randomization and multi-central protocol), previous treatment, number of patients per arm, and journal IF did not affect the endpoints. Type of the therapy (targeted or immunotherapy and chemotherapy) also did not affect the results.

### Discussion

We performed a meta-analysis of studies, based on phase II clinical trials and evaluated the main cancer treatment outcome metrics (RR, PFS, and OS). Personalization of cancer treatment, based on genomic data, improved the outcomes. The results were obtained with proportional meta-analysis and meta-regression and confirmed by non-parametric statistics.

**Table 2. Effects and parameters analysis.**

| Groups and Statistics | Parameters | RR | 1-year PFS rates | 1-year OS rates | PFS median | OS median |
|---|---|---|---|---|---|---|
| **Personalized** | Me | 30.1% | 41.3% | 100% | 26.59 | 9.9 |
| | Q1 | 17% | 30.2% | 48.8% | 11.92 | 7.25 |
| | Q3 | 45.6% | 56.7% | 100% | 41.04 | 13.6 |
| **Non-personalized** | Me | 12.1% | 27.9% | 69.4% | 16.65 | 6.7 |
| | Q1 | 9.4% | 19% | 39.3% | 10.43 | 4.8 |
| | Q3 | 17.3% | 38.8% | 89.2% | 21.15 | 9.69 |
| **M-W test Personalized vs Non-personalized, p-value** | | **0.009\*** | **0.011\*** | **0.058** | **0.018\*** | **0.032\*** |
| **Randomized** | Me | 43% | 32.5% | 80.4% | 19.15 | 7.8 |
| | Q1 | 28.9% | 27.9% | 50.8% | 12.5 | 6.69 |
| | Q3 | 45.6% | 44.4% | 100% | 27.15 | 10.65 |
| **Non-randomized** | Me | 43% | 34.2% | 72.9% | 17.5 | 8.35 |
| | Q1 | 28.9% | 24.6% | 44.3% | 10.62 | 6.15 |
| | Q3 | 45.6% | 49.4% | 100% | 33.54 | 11.57 |
| **M-W test Randomized vs Non-randomized, p-value** | | **0.746** | **0.887** | **0.983** | **0.817** | **0.889** |
| **Single-central protocol** | Me | 42.3% | 34.2% | 80.4% | 19.3 | 8 |
| | Q1 | 29.7% | 27.8% | 52.5% | 12.8 | 6.69 |
| | Q3 | 51.2% | 47.5% | 100% | 30.75 | 11.35 |
| **Multi-central protocol** | Me | 34% | 27.5% | 48.1% | 11.55 | 6.9 |
| | Q1 | 30% | 22.5% | 42.5% | 10.2 | 5.5 |
| | Q3 | 38.8% | 33.8% | 97.3% | 26.13 | 9.66 |
| **M-W test Single-central vs Multi-central, p-value** | | **0.279** | **0.207** | **0.401** | **0.33** | **0.502** |
| **Pre-treated** | Me | 31% | 34.2% | 62.9% | 15.10 | 8.6 |
| | Q1 | 26.6% | 17.9% | 33.6% | 8.08 | 4.55 |
| | Q3 | 39.4% | 41.7% | 100% | 27.63 | 10.23 |
| **Treatment naïve** | Me | 44.3% | 31.7% | 84.1% | 19.79 | 7.65 |
| | Q1 | 35.5% | 27.9% | 57.3% | 13.93 | 6.7 |
| | Q3 | 51.2% | 79.2% | 100% | 38.3 | 18.75 |
| **M-W test Pre-treated vs Treatment naïve, p-value** | | **0.037\*** | **0.274** | **0.077** | **0.36** | **0.059** |
| **Targeted/Immuno-therapy** | Me | 39.2% | 38.9% | 98.5% | 25.35 | 9.34 |
| | Q1 | 30% | 29.5% | 47.8% | 11.46 | 7.08 |
| | Q3 | 44.6% | 55.9% | 100% | 37.2 | 13.43 |
| **Chemotherapy** | Me | 38% | 32.5% | 75.6% | 18.57 | 7.95 |
| | Q1 | 28.9% | 22% | 42.5% | 11.09 | 5.38 |
| | Q3 | 68% | 49.4% | 100% | 28.95 | 11.57 |
| **M-W test Targeted/Immunotherapy vs Chemotherapy, p-value** | | **0.52** | **0.532** | **0.261** | **0.67** | **0.301** |
| **Number of patients per arm > 40** | Me | 37% | 30.6% | 73.7% | 17.23 | 7.4 |
| | Q1 | 29.8% | 24.1% | 46.1% | 11.07 | 5.95 |
| | Q3 | 44.9% | 42% | 100% | 25.49 | 10.05 |
| **Number of patients per arm ≤ 40** | Me | 43% | 35.8% | 80.8% | 19.39 | 9 |
| | Q1 | 31% | 29.4% | 54.4% | 13.05 | 7.1 |
| | Q3 | 50% | 96.4% | 100% | 38.6 | 26.75 |

*(Continued)*

**Table 2.** (Continued)

| Groups and Statistics | Parameters | RR | 1-year PFS rates | 1-year OS rates | PFS median | OS median |
|---|---|---|---|---|---|---|
| M-W test Number of patients per arm, p-value | | 0.655 | 0.123 | 0.42 | 0.096 | 0.306 |
| Journal 5 year IF > 10 | Me | 34% | 34.2% | 71.8% | 17.23 | 8.2 |
| | Q1 | 25.3% | 16.9% | 41% | 9.84 | 4.05 |
| | Q3 | 46.8% | 41.3% | 100% | 25.48 | 9.91 |
| Journal 5 year IF ≤ 10 | Me | 43% | 31.7% | 80.4% | 19.15 | 7.7 |
| | Q1 | 32% | 27.8% | 47.5% | 11.55 | 6.69 |
| | Q3 | 45.6% | 55% | 100% | 33.95 | 12.42 |
| M-W test Journal 5 year IF, p-value | | 0.235 | 0.363 | 0.321 | 0.313 | 0.33 |

*Significant difference, the Mann-Whitney (M-W) test, $p < 0.05$.

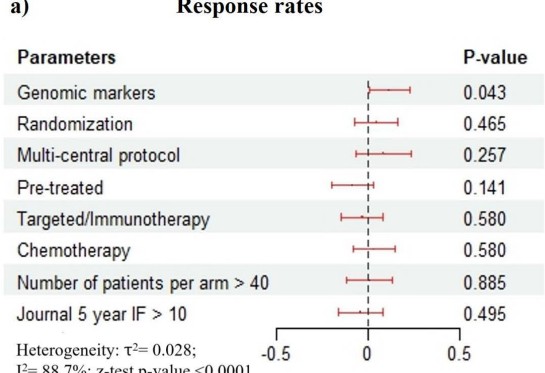

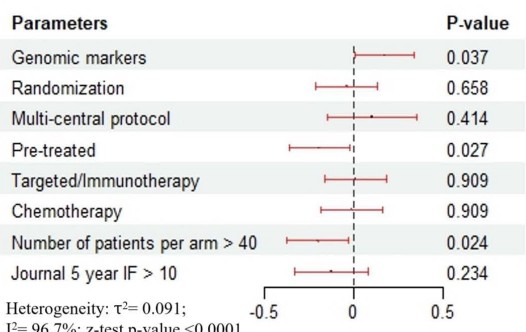

**Fig 2. Results of meta-regression. a)** RRs; **b)** 1-year PFS; **c)** 1-year OS. The plot shows the results of proportional meta-analysis and meta-regression. Meta-regression was performed between qualifying parameters and the size-effects of outcome metrics. Regression estimates and CIs are presented at the forest plots. Regression z-test p-values are presented in the right columns.

RR, 1-year, and medians of PFS and OS were significantly higher in studies, where the patients were selected on a genomics-based approach. Implementation of this approach improves the outcomes of any type of cancer drug treatment: cytotoxic chemotherapy, targeted therapy, and immunotherapy. The study design, journal IF, and number of patients per arm did not affect the outcomes. The results indicate the homogeneity of the studies and suggest a high strength of

**Table 3. 1-year PFS multivariable meta-regression.**

| Parameters \ Statistics | Regression estimate | P-value |
|---|---|---|
| Pre-treated | −0.163 | 0.048* |
| Number of patients per arm > 40 | −0.157 | 0.074 |
| Genomic markers | 0.124 | 0.135 |

Only parameters, significant in univariable analysis were included. * Significant difference, p < 0.05.

association between genomics-based personalization and better outcomes. 1-year survival rates indicate survivability on a certain observation time. The median survival rates indicate the general level considering study observation time, based on specific characteristics of diverse cancers [37]. Median meta-analysis without additional data is less accurate and powerful in comparison to proportional meta-analysis, therefore we compared medians only with the Mann-Witney test. We obtained significant differences between personalized and non-personalized studies that may indicate the strong importance of genomic tests for effective treatment strategy selection.

Schwaederle M. et al (2015) performed a similar meta-analysis of the effectiveness of a personal approach to cancer drug treatment [21]. The authors based their investigation on 2010–2012 data and received significantly better RR, PFS, and OS in the case of personalized studies. Only single-agent studies were analyzed, and personalization was considered from the general point of view. Next-generation sequencing, other biomarker tests, and a complex personal approach for patient selection were evaluated to distinguish personalized and non-personalized studies [21]. We performed a narrower search with more specific terms in PubMed but included multi-agent and immunotherapy arms. Immunotherapy is a crucial part of the treatment of various cancers, and a multi-agent approach is common in modern treatment strategies [38]. The searches included studies based on phase II clinical trial results, reporting trial protocol, and retrospective, based on subsequent evaluation of the trial data. We studied genomic agents as the type of personalization and included data obtained in 2012–2021, therefore our findings confirm the results from previous publications for other arm types [21] which was the novelty of the approach. Based on the previous experience, we confirmed the results and extended the research topic following newer cancer drug treatment standards.

Targeted therapy and immunotherapy mechanisms are developed against specific tumor phenotypes, associated with certain mutations [1,39,40]. At the same time, these types of therapy may be appointed without previous genetic tests, which leads to worse outcomes [40]. Cytotoxic chemotherapy is a system anti-cancer treatment, which is mostly not connected with specific tumor genotypes [41,42]. However, it is possible to suggest better outcomes of this treatment with previously evaluated genomic data. According to our results, prescriptions, based on genomic data, improve all types of cancer drug treatment outcomes. Sometimes there is no option in clinical practice guidelines to change the treatment strategy considering genetic mutations. It is possible to suggest genome profiling as a crucial part of diagnostics in all cases, where it is possible. This suggestion was described in a plethora of previous reviews and original articles [5,20,43]; the development of pharmacogenomic databases was discussed to be able to help in the development of a personal approach for cancer drug treatment that improves the general outcomes [44]. Our meta-analysis aimed to approve the suggestion from an evidence-based point of view.

Nowadays next-generation sequencing and other genomic technologies are included in standards of cancer diagnosis [44,45]. The rate of non-personalized therapy in published clinical trials decreased during the last decades from 95% to 34% [21]. Importantly, we first detected that non-personalized treatment (including targeted therapy, immunotherapy, chemotherapy, and multi-agent) led to worse 1-year outcomes.

IF was evaluated as an additional parameter to assess study variability and quality. The non-significant results of the meta-regression and subgroup analysis suggest comparable RR, OS, and PFS outcomes across studies published in

both high-impact and other journals. These findings indicate that the ability of genomics-based personalization to improve treatment outcomes is consistently reported, regardless of journal impact.

The lack of consensus regarding the optimal implementation of precision therapy underscores the significance of conducting a meta-analysis in this domain. Further research endeavors in this realm hold the potential to diminish skepticism and apprehension surrounding the adoption of precision medicine modalities. Many medical professionals and researchers harbor uncertainties concerning the advantages of precision medicine in improving clinical outcomes, especially considering the augmented costs associated with individualized therapy [46]. Precision medicine demonstrates the capacity to refine various healthcare strategies leading to more economically efficient treatment protocols, better treatment outcomes, and heightened quality of life for cancer patients. Appreciating the repercussions of personalized treatment strategies can facilitate a more rigorous approach to clinical decision-making and foster advancements in patient outcomes [4]. Our study uncovers global evidence for the positive impact of genomics-based personalization and provides the ability to disseminate the general benefits of a precision approach for cancer treatment which is important for personal oncology development [19]. Our results facilitate wider implementation of targeted and immuno-oncological drugs in the modern world including developing and low-income countries, where these treatments remain poorly accessible to patients [13,19].

### Study novelty and limitations

This study presents a novel analysis of outcomes of a precision approach for cancer drug treatment selection. We did not find any similar studies since 2015 [47], so the previous results were updated and upgraded, considering larger and novel publication dates from 2010 to 2021. The extended period allowed us to evaluate actual targeted, immunotherapy, and multi-agent (considered targeted/immunotherapy) phase II clinical trials. We first simultaneously include in the analysis targeted, immuno-oncological, chemotherapeutic, and multi-agent studies, uncovering the evidence for the effectiveness of genomics-based cancer treatment personalization. The novelty of the study also includes the implementation of a proportional meta-analysis for 1-year OS and PFS evaluation.

This study describes a global trend, based on meta-analysis of specific studies. We included 50 diverse studies, having 81 arms, so for more detailed results, various articles should be evaluated based on cancer type, specific therapies, employed genomic technologies, and studied mutations. Only Phase II clinical trial data collected from the PubMed database is a limitation of the study. The studies varied in genomic markers and demonstrated a high level of heterogeneity across RR, OS, and PFS. Oncological diseases are quite heterogeneous in general; RR, OS, and PFS depend on cancer type, disease stage, and treatment. Future analysis may include Phase III clinical trials, collected from various databases with an extended subgroup analysis and an assessment of cost-effectiveness and quality-of-life outcomes. Evaluation of calculated 1-year survival rates did not allow to employ diverse meta-analysis models in one study. Based on genomic data, we approved personalization as an effective way for precision oncology development and to improve cancer treatment outcomes [48].

In conclusion, we performed a novel meta-analysis of current Phase II clinical trial results, extracted from various studies. Our results approving the effectiveness of the precision approach for cancer drug treatment selection are important for cancer healthcare development and personalization. Further prospective multi-central clinical trials are in demand to provide additional robust evidence for each type of cancer.

### Supporting information

**S1 Checklist.  PRISMA checklist.**
(PDF)

**S2 Table.  RoB assessment of study quality.**
(PDF)

**S3 Table. Assessment of study quality.** Grade approach.
(PDF)

**S4 Dataset. Data extracted from included studies and arms** https://www.doi.org/10.5281/zenodo.10966214.
(XLSX)

**S5 File. Description of 803 excluded and included studies with exclusion reasons** https://www.doi.org/10.5281/zenodo.10966214.
(XLSX)

**S6 File. Code of the analysis.** https://github.com/MikhailPot/PreciseOnco_meta-analysis.
(PDF)

## Author contributions

**Conceptualization:** Mikhail B. Potievskiy, Peter V. Shegai.

**Data curation:** Elena P. Zharova, Lidia A. Nekrasova, Airat I. Garifullin, Ivan V. Korobov, Nikita E. Shevchenko.

**Formal analysis:** Mikhail B. Potievskiy, Lidia A. Nekrasova, Airat I. Garifullin, Ivan V. Korobov, Nikita E. Shevchenko.

**Investigation:** Mikhail B. Potievskiy, Lidia A. Nekrasova.

**Methodology:** Mikhail B. Potievskiy.

**Project administration:** Mikhail B. Potievskiy, Andrei D. Kaprin, Peter V. Shegai.

**Supervision:** Peter V. Shegai.

**Validation:** Anastasia A. Zabolotneva, Andrei D. Kaprin, Peter V. Shegai.

**Writing – original draft:** Mikhail B. Potievskiy, Lidia A. Nekrasova.

**Writing – review & editing:** Mikhail B. Potievskiy, Elena P. Zharova, Anastasia A. Zabolotneva, Dmitriy N. Atochin, Andrei D. Kaprin, Peter V. Shegai.

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
