## [Decision Letter · Decision Letter 0]

23 Jan 2025

PONE-D-24-46559Personal Approach for Cancer Treatment: a Meta-Analysis of Phase II Clinical TrialsPLOS ONE

Dear Dr. Potievskiy,

Thank you for submitting your manuscript to PLOS ONE. After careful consideration, we feel that it has merit but does not fully meet PLOS ONE’s publication criteria as it currently stands. Therefore, we invite you to submit a revised version of the manuscript that addresses the points raised during the review process.

We look forward to receiving your revised manuscript.

Kind regards,

Norikatsu Miyoshi, M.D., Ph.D., F.A.C.S., F.A.S.C.R.S.,

Academic Editor

PLOS ONE

Journal Requirements:

2. As required by our policy on Data Availability, please ensure your manuscript or supplementary information includes the following: 

Reviewers' comments:

Reviewer's Responses to Questions

**Comments to the Author**

1. Is the manuscript technically sound, and do the data support the conclusions?

Reviewer #1: Yes

Reviewer #2: Partly

2. Has the statistical analysis been performed appropriately and rigorously? 

Reviewer #1: Yes

Reviewer #2: I Don't Know

3. Have the authors made all data underlying the findings in their manuscript fully available?

Reviewer #1: Yes

Reviewer #2: Yes

4. Is the manuscript presented in an intelligible fashion and written in standard English?

Reviewer #1: Yes

Reviewer #2: Yes

5. Review Comments to the Author

Reviewer #1: The manuscript provides a comprehensive and well-structured analysis of the impact of personalized genomics-based cancer treatments compared to non-personalized approaches. The combination of a systematic review and meta-analysis, grounded in PRISMA guidelines, enhances the robustness and transparency of the findings. The data-driven approach, including proportional meta-analysis and meta-regression, supports the study's conclusions about the significant benefits of personalized therapy in improving key clinical outcomes such as response rate, progression-free survival, and overall survival.

The inclusion of 46 studies spanning 81 arms and over 6,500 patients demonstrates the extensive effort to capture a wide range of clinical trials and provides a strong foundation for the conclusions drawn. The significant positive results, particularly the association of personalized treatment with improved response rates and survival metrics, are timely and relevant for advancing oncology research and clinical practice.

Moreover, the meticulous methodology, including independent data extraction and cross-checking by multiple reviewers, adds credibility to the results. The article highlights the promise of genomics-based approaches for tailoring cancer therapies, underscoring their potential for enhancing patient outcomes.

Minor Revisions Suggested

Clarify the methodology regarding the selection criteria for included studies, particularly how studies with mixed treatment types were handled.

Expand on the limitations of the study, especially concerning the heterogeneity of included trials and the potential variability in genetic markers across studies.

Consider a brief discussion on the implications of journal impact factor, even if found non-significant, as this adds context to the study design evaluation.

Include additional details on the statistical tests used (e.g., assumptions for the Mann-Whitney test) to ensure reproducibility.

In summary, this work makes a valuable contribution to the field of oncology by providing compelling evidence for the clinical benefits of personalization in cancer treatment. Upon addressing the minor revisions suggested, I strongly recommend its acceptance for publication, as it will likely inspire further research and have a meaningful impact on both clinical and research communities.

Reviewer #2: This study presents a meta-analysis of Phase II clinical trials, showing that genomics-based personalized cancer treatments significantly enhance response rates, progression-free survival, and overall survival compared to non-personalized therapies. To make the article more informative, we suggest addressing the following points:

First, the study reports considerable heterogeneity across key outcomes (RR, PFS, OS). Conducting subgroup analyses (e.g., by cancer type, treatment modality, or geographical region) and sensitivity analyses could help identify and address potential sources of heterogeneity.

Second, the data collection is restricted to PubMed, which may lead to the exclusion of relevant studies. Expanding the search to include additional databases, such as EMBASE or Cochrane Library, would ensure a more comprehensive literature review.

Lastly, as the analysis focuses exclusively on Phase II trials, the clinical applicability of the findings may be limited. Including Phase III trial data or real-world evidence, along with an assessment of cost-effectiveness and quality-of-life outcomes, could significantly enhance the practical relevance and impact of the study's conclusions.

6. PLOS authors have the option to publish the peer review history of their article (what does this mean? ). If published, this will include your full peer review and any attached files.

**Do you want your identity to be public for this peer review?** For information about this choice, including consent withdrawal, please see our Privacy Policy .

Reviewer #1: **Yes: ** Mohamed ABDELKARIM

Reviewer #2: No

---

## [Author Response · Author response to Decision Letter 1]

5 Apr 2025

We thank the reviewers and the editor for the crucial comments that will improve our manuscript.

The supplements were revised and corrected following the journal’s data availability policy. We added additional information on the Risk of bias evaluation and provided the reasons for exclusion for all the selected trials. We also added supportive information on the data extraction process (names of the reviewers and date of extraction). The dataset and the information on excluded and included studies were revised and corrected following the journal's requirements (S4 Dataset, S5 File https://www.doi.org/10.5281/zenodo.10966214). The results of the certainty of evidence assessment (GRADE approach) are provided in S3 Table, and the method description is included on p.7.

Please find below our detailed responses to each point raised by reviewers.

Responses to Reviewer 1.

Reviewer 1 commented: “Clarify the methodology regarding the selection criteria for included studies, particularly how studies with mixed treatment types were handled”.

Response: We added additional information in the Methodology section to clarify the selection criteria, especially for mixed studies: The corrections were made in the 1st paragraphs of the “Eligibility Criteria and Study Selection Strategy” section on p.5 (“Studies of chemotherapy, immunotherapy, targeted therapy, and multi-agent trials, including various treatments, were selected.”) and the “Data Synthesis and Statistics” section on p.7 (“We considered multi-agent therapy as targeted or immune-oncological treatment if those were appointed in combination with other treatments.”).

Reviewer 1 suggested: “Expand on the limitations of the study, especially concerning the heterogeneity of included trials and the potential variability in genetic markers across studies”.

Response: The study limitations section was extended with a focus on study heterogeneity on page 15 (Only phase II clinical trial data collected from the PubMed database is a limitation of the study. The studies varied in genomic markers and demonstrated a high level of heterogeneity across RR, OS, and PFS. Oncological diseases are quite heterogeneous in general; RR, OS, and PFS depend on cancer type, disease stage, and treatment. Future analysis may include Phase III clinical trials, collected from various databases with an extended subgroup analysis and an assessment of cost-effectiveness and quality-of-life outcomes.”).

Reviewer 1: “Consider a brief discussion on the implications of journal impact factor, even if found non-significant, as this adds context to the study design evaluation”.

Response: We included a discussion on the implications of the journal impact factor on page 14 (IF was evaluated as an additional parameter to assess study variability and quality. The non-significant results of the meta-regression and subgroup analysis suggest comparable RR, OS, and PFS outcomes across studies published in both high-impact and other journals. These findings indicate that the ability of genomics-based personalization to improve treatment outcomes is consistently reported, regardless of journal impact.”.

Reviewer 1 commented: “Include additional details on the statistical tests used (e.g., assumptions for the Mann-Whitney test) to ensure reproducibility”.

Response: We added additional information on the statistical tests and clarified the conditions for their use on page 7 (“We compared the effects using the Chi-square test for categorical variables and employed the t-test or Mann-Whitney test based on the type of distribution (normal or non-normal) for continuous variables (RR, OS, and PFS), with a significance level set at p<0.05.”).

Responses to Reviewer 2

Reviewer 2 commented: “First, the study reports considerable heterogeneity across key outcomes (RR, PFS, OS). Conducting subgroup analyses (e.g., by cancer type, treatment modality, or geographical region) and sensitivity analyses could help identify and address potential sources of heterogeneity.”

Response: We agreed that high heterogeneity is a limitation of the study and included several sentences to highlight it in the 2nd paragraph of page 15 (“The studies varied in genomic markers and demonstrated a high level of heterogeneity across RR, OS, and PFS.”). Thank you for the suggestion that potential reasons for such results may be associated with various cancer types and treatments. Oncological diseases are quite heterogeneous in general; RR, OS, and PFS depend on cancer type, disease stage, and treatment. We described this topic in the discussion section, extending the limitations on page 15 (“Oncological diseases are quite heterogeneous in general; RR, OS, and PFS depend on cancer type, disease stage, and treatment.”). We also enhanced the explanation of heterogeneity on page 10 (“High heterogeneity rates may be associated with different response absolute metrics of outcomes for various cancers, disease stages, and treatments.”).

Reviewer 2 commented: “Second, the data collection is restricted to PubMed, which may lead to the exclusion of relevant studies. Expanding the search to include additional databases, such as EMBASE or Cochrane Library, would ensure a more comprehensive literature review.”

Response: We underlined such study limitations as PubMed and Phase II clinical trial restricted data search on page 15 (“Only Phase II clinical trial data collected from the PubMed database is a limitation of the study.”).

Reviewer 2 commented: “Lastly, as the analysis focuses exclusively on Phase II trials, the clinical applicability of the findings may be limited. Including Phase III trial data or real-world evidence, along with an assessment of cost-effectiveness and quality-of-life outcomes, could significantly enhance the practical relevance and impact of the study's conclusions”.

Response: We agree that the recommendations provided will make the study more informative and impactful, however, it will require reorganization of the manuscript after full re-analysis in accordance with the PRISMA workflow. This manuscript is the first step in genomics-based personalization analysis, and our current research includes the extended analysis suggested by the reviewer. A consideration of future research was included in the revised version of the manuscript on page 15 (“Future analysis may include Phase III clinical trials, collected from various databases with an extended subgroup analysis and an assessment of cost-effectiveness and quality-of-life outcomes.”).

---

## [Editor Report · Decision Letter 1]

2 Sep 2025

Personal Approach for Cancer Treatment: a Meta-Analysis of Phase II Clinical Trials

PONE-D-24-46559R1

Dear Dr. Potievskiy,

We’re pleased to inform you that your manuscript has been judged scientifically suitable for publication and will be formally accepted for publication once it meets all outstanding technical requirements.

Kind regards,

Satyajeet Rath

Academic Editor

PLOS ONE

Additional Editor Comments:

Thank you for making the desired corrections. We are happy to inform you that the article have been accepted for publication.

---

## [Editor Report · Acceptance letter]

PONE-D-24-46559R1

PLOS ONE

Dear Dr. Potievskiy,

I'm pleased to inform you that your manuscript has been deemed suitable for publication in PLOS ONE. Congratulations! Your manuscript is now being handed over to our production team.

Kind regards,

on behalf of

Dr. Satyajeet Rath

Academic Editor

PLOS ONE